# Intraoperative Prediction of Coronary Graft Failure Based on Transit Time Flow Measurement: A PRELIMINARY STUDY

**DOI:** 10.3390/diagnostics14171903

**Published:** 2024-08-29

**Authors:** Boris N. Kozlov, Vasily V. Zatolokin, Andrew V. Mochula, Yusufjon Alisherov, Dmitri S. Panfilov, Nikolay O. Kamenshchikov, Elena B. Kim

**Affiliations:** Cardiology Research Institute, Tomsk National Research Medical Center, Russian Academy of Sciences, 111a Kievskaya St., Tomsk 634012, Russia; bnkozlov@yandex.ru (B.N.K.); zatolokin@cardio-tomsk.ru (V.V.Z.); mochula@cardio-tomsk.ru (A.V.M.); lion19-93@inbox.ru (Y.A.); pand2006@yandex.ru (D.S.P.); nikolajkamenof@mail.ru (N.O.K.)

**Keywords:** transit time flow measurement, graft patency, coronary flow reserve, coronary bypass grafting, coronary bypass flow reserve

## Abstract

Myocardial revascularization has been known to not affect the prognosis in some patients. Coronary artery bypass graft (CABG) failure may develop one year after CABG surgery. This is accompanied by a high risk of developing myocardial infarction after complete myocardial revascularization in obstructive coronary artery disease (CAD) due to microvascular dysfunction. The study of microvascular dysfunction using intraoperative stress tests with adenosine triphosphate (ATP) allows for the assessment of the coronary bypass flow reserve (CBFR) and the risk of graft failure one year after surgery. The study included 79 CAD patients (238 grafts) who underwent dynamic single-photon emission computed tomography (SPECT) before CABG and dynamic transit time flow measurement (TTFM) during CABG at rest and at stress. The CBFR was calculated by the ratio of the mean graft flow (MGF) at stress to the MGF at rest. A multivariate regression model showed that the MGF at rest (*p* = 0.043), the MGF at stress (*p* = 0.026) and the CBFR (*p* = 0.0001) were significant independent predictors of graft failure. As a result of ROC analysis, the threshold CBFR < 1.67 units correlated with graft failure more closely (sensitivity 82%, specificity 90%) The CBFR is a significant independent predictor of graft failure for up to 16 months.

## 1. Introduction

Coronary artery bypass grafting (CABG) is an effective treatment for patients with coronary artery disease (CAD). The success of CABG surgery, as well as the long-term patency of coronary bypass grafts, depends not only on the quality and type of graft chosen but also on the condition of the coronary bed [1,2]. The patency of coronary artery bypass grafts after surgery is a major determinant of morbidity and mortality after surgical myocardial revascularization. However, according to the literature, after CABG, mammary–coronary graft failure is detected in up to 8% of cases, and the great saphenous vein graft failure develops in up to 20% of cases during the first year of follow-up [1]. In the long-term period after surgery, graft failure is associated with intimal hyperplasia in autovenous grafts and the progression of atherosclerosis in native coronary arteries. As a rule, early postoperative graft failure is caused by a surgical error (graft twisting, linear tension due to insufficient length and stenosis in the anastomotic area), which can be corrected only during surgery. Another reason for early graft failure may be a distal coronary bed failure caused by microcirculatory dysfunction that contributes to the onset of microvascular angina, which is diagnosed when computed tomography angiography shows a reduced coronary flow reserve (CFR) (less than 2.0 units) [3]. The treatment of microvascular angina is aimed at relieving the dominant mechanisms of microcirculatory dysfunction, and therefore, an optimal therapeutic treatment regimen is recommended, which differs from the treatment regimen for patients with isolated obstructive coronary pathology [3].

To date, the method of choice for assessing coronary microvascular function is positron emission tomography (PET). PET allows for measuring absolute myocardial blood flow (MBF) in mL/min/g of tissue, as well as for evaluating the CFR. The high informative value of this method has been noted in a number of large trials [4]. An alternative method for assessing coronary microcirculation can be considered dynamic single-photon emission computed tomography (SPECT). This technique is implemented with new CZT gamma cameras. It measures both the MBF and CFR. Some researchers devoted their work to the validation of the dynamic SPECT of the myocardium [4] At the same time, compared to dynamic SPECT, PET is characterized as challenging and high cost, which makes it unavailable for widespread use in clinical practice.

Currently, the most informative and safe way to intraoperatively assess graft patency is transit time flow measurement (TTFM), which is an intraoperative ultrasound flow measurement [5]. The TTFM findings closely correlate with the data of early postoperative angiography; therefore, the predictive value of this technique is high [6]. In the vast majority of cases, TTFM is performed once during surgery against the background of restoration of stable hemodynamics [7]. This approach cannot be considered ideal since a single measurement of blood flow at the end of the main stage of the operation does not allow for the assessment of the hemodynamic functional state of the coronary microcirculation. Intraoperative TTFM in bypass grafts under pharmacological stress demonstrates the coronary bypass flow reserve (CBFR) based on the condition and reactivity of the coronary microvasculature in patients immediately after surgery. Measuring the CBFR at the end of the operation immediately after complete myocardial revascularization makes it possible to diagnose microcirculatory coronary dysfunction that worsens the patency of newly formed coronary bypass grafts. This flow measurement of CBFR will allow for immediate correction of the optimal drug treatment of patients and the clinical examination schedule in the postoperative period.

The aim of the study was to assess the risk of coronary artery bypass graft failure depending on the threshold value of the CBFR during surgical myocardial revascularization using the TTFM.

## 2. Materials and Methods

The study included 79 patients (238 grafts) with CAD. All patients preoperatively underwent cardiac dynamic SPECT according to a two-day protocol, at functional rest and against the background of a pharmacological stress test using a Discovery NM/CT570C scanner (GE Healthcare, Chicago, IL, USA). At the first stage, the passage of a bolus of Technetium, 99mTc radiopharmaceutical (RP) through the chambers and LV myocardium at functional rest was recorded. The second stage of cardiac dynamic SPECT was carried out the next day and included the recording of RP bolus passage through the chambers and LV myocardium, against the background of a pharmacological test with adenosine triphosphate (ATP), which was administered intravenously at a dose of 160 μg/kg/min.

All patients underwent CABG surgery using the internal mammary artery to the left anterior descending artery and the great saphenous vein to other coronary arteries. All CABG surgeries were performed through the median sternotomy using cardiopulmonary bypass (CPB) and cardioplegia with full heparinization (initial heparin dose was 3 mg/kg). During CABG, after weaning from CPB, the TTFM was performed at rest and under pharmacological stress using the VeriQ System TTFM unit (Medistim, Oslo, Norway). This system allows for TTFM to use sensors of the appropriate size (2, 3, 4 and 5 mm). The main variables for assessing the CFR at rest were the mean graft flow (MGF), the pulsatility index (PI) and diastolic filling (DF). The indication for bypass graft revision was PI < 5 units. The pharmacological stress test was carried out using a metabolic adenosine triphosphate (ATP) in order to calculate the CFR. It was calculated by the ratio of the MGF under pharmacological stress to the MGF at rest. The required rate of ATP administration (160 μg/kg/min) was calculated individually, depending on the patient’s weight in accordance with the guidelines for conducting pharmacological stress tests [8]. ATP was administered intravenously using a mechanical infusion pump for 4 min. At the peak of the pharmacological load, at the end of the 4th mini-infusion of ATP, the TTFM was measured and recorded.

All patients underwent controlled multislice spiral computed tomography coronary angiography on average 16 ± 2 months after surgery.

### Statistical Analysis

The normality of the distribution was assessed with the Kolmogorov–Smirnov test. The results were expressed using the median and interquartile range. The comparisons were made with the Mann–Whitney U test for non-normal distributions. Initially, a univariate logistic regression analysis was carried out for each of the potential predictor variables of graft failure. Variables significant at *p* < 0.05 were also included in a multivariate regression model. The ROC analysis was used in this study. Statistical analysis was performed with the Statistical Package for the Social Sciences (SPSS) Version 11.0 statistic software package. A *p* value < 0.05 was considered statistically significant.

## 3. Results

Table 1 presents patient demographics.

The MBF was assessed according to SPECT in 238 coronary arteries of 79 patients with multivessel CAD preoperatively. The mean rest single-photon emission computed tomography (RS SPECT) was 0.63 ± 0.36 mL/min/g. During the stress load with ATP, the MBF increased to 0.9 ± 0.44 mL/min/g (Table 1). Thus, the CFR in patients was 1.59 ± 0.73 units (Table 2).

All patients underwent CABG using the left internal mammary artery (LIMA) for the left anterior descending (LAD) artery. All other coronary arteries were grafted with the great saphenous vein. All measurements were performed using the VeriQ System TTFM unit. During TTFM at rest, we did not obtain PI < 3 units; therefore, no revision of the anastomoses was performed. During surgery, the CBFR was measured in all grafts using dynamic TTFM. The MGF at rest was 49 ± 14 mL/min; at stress loading, the MGF increased to 97.8 ± 31 mL/min. The CBFR was 1.96 ± 0.43 units (Table 3). The DF and PI averaged between 68 ± 11.6% and 2.3 ± 1.2 units, respectively (Table 3).

According to multislice spiral computed tomography coronary angiography, the graft patency was 96.2% after 16 ± 2 months. A multivariate regression model showed that the MGF at rest (*p* = 0.043), the MGF at stress (*p* = 0.026) and the CBFR (*p* = 0.0001) were significant independent predictors of early graft failure (Figure 1) (Table 4). The coordinates of the curve demonstrate the threshold value of the CBFR, which correlated with early graft failure more closely in cases where it was less than 1.67 units (sensitivity 82%, specificity 100%) (Table 3). As a result of multivariate regression analysis, more frequent coronary bypass graft failure was observed after 16 ± 2 months at a threshold of intraoperative CBFR < 1.67 units (sensitivity 82%, specificity 100%) (Figure 1) (Table 5).

### Study Limitations

The subgroup analysis of patients with baseline CBFR < 1.69 may not be robust due to the relatively small sample size. Our study lacks any observation or analysis of the group receiving treatment for microvascular dysfunctions. Therefore, the relevance and value of CBGF diagnosis need to be assessed in the future when applying the optimal method for treating microvascular dysfunctions.

## 4. Discussion

Currently, there is a lack of attention being paid to microvascular dysfunction of the coronary bed in the surgical treatment of obstructive CAD. After all, obstructive lesions in coronary arteries do not exclude the risk of simultaneous microvascular angiopathy, which affects the efficacy of grafts both during surgery and in the postoperative period. Both in the scientific literature and in modern guidelines for the treatment of chronic coronary syndrome, the diagnostic search for microcirculatory disorders, namely the CFR, is initiated only in conditions of non-obstructive, hemodynamically insignificant lesions in coronary arteries [3]. The CFR is known to be reduced both due to obstructive CAD and microcirculatory dysfunction [8,9,10]. Myocardial revascularization increases myocardial flow reserve in the majority of patients with multivessel CAD, whose baseline CFR was reduced [9,10]. Nevertheless, there are studies that report a persistently reduced CFR after myocardial revascularization, which is accompanied by a higher risk of ischemic events and coronary artery bypass graft failure in the postoperative period, most likely due to not only coronary artery obstruction in such patients but also microcirculatory disorders that require special attention [8].

Intraoperative TTFM is known to be the most modern method of intraoperative quality control of CABG [11,12]. However, ultrasound flow measurement is currently used only to assess the quality of formed anastomosis, and in most studies, parameters such as an MGF of less than 15 mL/min and a PI of more than 3 units are considered indications for the revision of coronary bypass grafts [12]. It is also worth noting that in most cases, ultrasound flow measurement is performed only once during surgery after weaning from CPB [5,6,7,11]. We suppose that this strategy of single ultrasound flow measurement does not reveal all the possibilities of intraoperative TTFM. Much more information can be obtained when it is used repeatedly during surgery, for instance, immediately after the distal anastomosis completion on an arrested heart (on cross-clamp), as well as the proximal anastomosis creation during ongoing CPB, after weaning from CPB and after protamine administration when the chest retractor has been removed (before the chest closure), which makes it possible to detect a technical error and correct it immediately [13,14]. It is worth noting that some authors suggest conducting pharmacological stress tests to make a decision on the revision of graft anastomosis with MGF [12]. However, the most informative and predictive of poor anastomosis is a flow parameter such as the PI, and its value of more than 3 units is considered a generally accepted indicator for the revision of anastomosis [12]. The volumetric blood flow measurement at a PI of less than 3 units cannot be an indicator for the revision of anastomosis, since it reflects only the volume of blood flow that the distal channel can receive in each specific case under physiological rest [12,15]. Therefore, conducting pharmacological stress tests makes it possible to intraoperatively assess the dynamic graft patency with maximum microcirculatory vasodilation, which in turn reflects the CBFR, which has never been discussed in the literature before.

In the present study, we first analyzed the CFR according to dynamic SPECT in patients before CABG in order to assess the parameters of microcirculatory coronary flow against the background of multivessel hemodynamically significant lesions of the coronary arteries. Our study confirmed the low CFR, which was 1.59 ± 0.73 units according to dynamic SPECT, in patients with multivessel CAD in the preoperative period.

Next, in our study, during CABG surgery, each patient underwent TTFM after weaning from CPB to evaluate the function of the coronary bypass grafts at rest. The average MGF at rest was 49 ± 14 mL/min with DF at 68 ± 11.6% and a PI of 2.3 ± 1.2 units. Taking into account a PI of less than 3 units, the revision of coronary bypass anastomosis was not performed. Subsequently, all patients underwent a pharmacological stress test (ATP) with subsequent TTFM. During stress, the MGF increased on average to 97.8 ± 31 mL/min. For each patient, CBFR was calculated using the MGF stress/MGF, which averaged 1.96 ± 0.43 units.

The graft patency was 96.2% after 16 ± 2 months, according to multislice spiral computed tomography coronary angiography. The ROC analysis demonstrated a statistically significant dependence of the graft patency on the MGF at rest, the MGF at stress and the CBFR. But the most significant dependence was demonstrated by the CBFR (*p* = 0.0001). Our study found that when the CBFR was less than 1.67 units (sensitivity 82%, specificity 100%), it correlated with postoperative (up to 16 months) graft failure according to coronary angiography.

As a result of our study, it was found that some patients with multivessel CAD have a risk of having microcirculatory disorders, which are not possible to identify in the preoperative period. Dynamic intraoperative ultrasound flow measurement makes it possible to identify coronary grafts with low CBFR, which makes it possible to predict its early failure not due to poorly formed anastomosis, but as a result of microcirculatory disorders, the verification of which has now become possible thanks to ultrasound intraoperative flow measurement.

In world practice, there is drug therapy with fairly effective results in the treatment of microcirculatory disorders, including such drugs as NO donors and ATP-sensitive potassium channel activators (Nicorandil), NO modulators (Sildenafil) and others [16]. It is possible that in the context of a changing profile of patients (older age, a higher number of comorbidities, with multiple PCIs, etc.) aimed at CABG surgery, optimal drug therapy requires special attention and modernization to ensure longer patency of coronary bypass grafts.

## 5. Conclusions

Myocardial revascularization has no effect on prediction in some patients, as graft failure and persistent angina may develop one year after CABG. This suggests that additional diagnostic measures are required in patients after surgical myocardial revascularization to identify microvascular dysfunctions that affect both the patency of grafts and the course of CAD in general. Patients with CBGF less than 1.67 require special monitoring, a medical examination schedule and an optimal therapeutic treatment regimen.

## Figures and Tables

**Figure 1 diagnostics-14-01903-f001:**
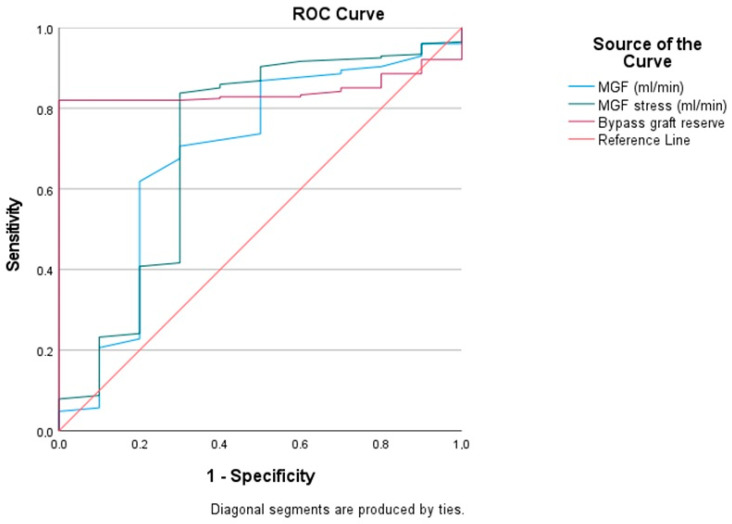
The ROC analysis. NB: MGF—mean graft flow at rest; MGF stress—mean graft flow at stress. Legend: The ROC analysis represents the cut-off CBGF values for predicting early graft failure. Cut-off −1.69; AUC (area under the curve) −0.844; 95% CI: (confidence interval) −0.79–0.89.

**Table 1 diagnostics-14-01903-t001:** Patient demographics.

Variable	*n* = 78
Age, years	60.0 (53.0, −68.0)
Female	28 (35.9%)
BMI, kg/m^2^	29.8 (25.6, −34.0)
Hypertension, *n* (%)	61 (78.2%)
Diabetes mellitus, *n* (%)	19 (24.3%)
Dyslipidemia	58 (74.3%)
Previous myocardial infarction	47(60.3%)
Family history of CVD	39 (50%)
Current smoker	41 (52.6%)
Previous stroke or TIA	13 (16.6%)

NB: BMI—body mass index, TIA—transient ischemic attack, CVD—cardiovascular disease.

**Table 2 diagnostics-14-01903-t002:** MBF and CFR according to dynamic SPECT data.

	Minimum	Maximum	Mean	Std. Deviation
RS SPECT (mL/min/g)	0.14	1.93	0.63	0.36
SS SPECT (mL/min/g)	0.12	2.45	0.9	0.44
CFR SPECT	0.35	5.52	1.59	0.73

NB: RS SPECT—rest single-photon emission computed tomography; SS SPECT—stress single-photon emission computed tomography; CFR SPECT—coronary flow reserve single-photon emission computed tomography.

**Table 3 diagnostics-14-01903-t003:** Dynamic TTFM data.

	Minimum	Maximum	Mean	Std. Deviation
MGF (mL/min)	21	102	49.62	14.33
MGF stress (mL/min)	31	198	97.79	31.44
CBFR	0.64	3.50	1.97	0.43
PI	1.0	4.8	2.27	1.19
DF	1.71	89.00	68.28	11.69

NB: MGF—mean graft flow at rest; MGF stress—mean graft flow at stress; CBFR—coronary bypass flow reserve; PI—pulsatility index; DF—diastolic filling.

**Table 4 diagnostics-14-01903-t004:** ROC analysis. The area under the curve.

Test Result Variable(s)	AUC Area	Asymptotic Sig. *	Asymptotic 95% Confidence Interval	
			Lower Bound	Upper Bound
Coronary artery stenosis	0.39	0.24	0.21	0.57
RS SPECT (mL/min/g)	0.46	0.69	0.28	0.64
SS SPECT (mL/min/g)	0.47	0.77	0.26	0.68
CFR SPECT	0.49	0.89	0.32	0.66
MGF (mL/min)	0.69	0.04	0.50	0.87
MGF stress (mL/min)	0.70	0.02	0.51	0.90
CBFR	0.84	0.00	0.79	0.89
PI	0.48	0.82	0.30	0.65
DF (%)	0.58	0.40	0.42	0.73

NB: AUC—area under the curve, RS SPECT—rest single-photon emission computed tomography, SS SPECT—stress single-photon emission computed tomography, MGF—mean graft flow, MGF stress—mean graft flow at stress, CBFR—coronary bypass flow reserve, PI—pulsatility index, DF—diastolic filling. The test variables have at least one tie between the positive actual state group and the negative actual state group. Statistics may be biased. * Null hypothesis: true area = 0.5.

**Table 5 diagnostics-14-01903-t005:** ROC analysis. Criterion values and coordinates of the ROC curve.

Criterion	Sensitivity	95% CI	Specificity	95% CI
≥0.64	100.00	98.4–100.0	0.00	0.0–30.8
>1.38	92.11	87.8–95.3	0.00	0.0–30.8
>1.39	92.11	87.8–95.3	10.00	0.3–44.5
>1.49	88.60	83.7–92.4	10.00	0.3–44.5
>1.5	88.60	83.7–92.4	20.00	2.5–55.6
>1.58	85.09	79.8–89.4	20.00	2.5–55.6
>1.59	85.09	79.8–89.4	30.00	6.7–65.2
>1.6	82.89	77.4–87.5	40.00	12.2–73.8
>1.62	82.89	77.4–87.5	60.00	26.2–87.8
>1.65	82.02	76.4–86.8	70.00	34.8–93.3
**>1.69**	**82.02**	**76.4**–**86.8**	**100.00**	**69.2–100.0**

## Data Availability

These study data are available from the corresponding author upon reasonable request.

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
