# Peer review of "Intraoperative Prediction of Coronary Graft Failure Based on Transit Time Flow Measurement: A PRELIMINARY STUDY"

_diagnostics, 2024, doi:10.3390/diagnostics14171903_

Round 1
Reviewer 1 Report
Comments and Suggestions for Authors
The fact that low level of coronary bypass flow reserve (CBFR) should be a most informative predictor of graft failure seems to be no wonder. However, the experimental confirmation of this self-evident effect is of independent value. The authors demonstrated that the level of CBRF which may be obtained during the operation permits assessment of the possibility of graft failure due to microvascular disorders. This knowledge may be very useful for further treatment of operated patients.
However, when reading the manuscript, a number of questions and, accordingly, critical remarks arise.
Firstly, to assess coronary reserve, the authors administered intravenous ATP, which relaxes smooth muscle in an endothelium-dependent manner. Naturally, the resulting increase in coronary blood flow is determined not only by the tone and patency of microvessels, but also by the state of the endothelium of the entire vascular bed, including relatively large coronary arteries. It is unclear how to distinguish the ability to relax the smooth muscles of microvessels against this background.
(Minor point: I should also note here that 160 ug/kg/min is not a dose, but a rate of administration, which can be converted into a dose when multiplied by the infusion time (4 min).)
Secondly, in Tables 1 and 2, the number of patients (N=238) should be removed to the legend, since the appearance of this number in the tables violates the dimensions of the rows and is completely inappropriate.
In these tables, it is also necessary to analyze the numbers that characterize mean and standard deviation. These values obtained during calculations cannot have so many significant digits after the decimal point. This meaninglessness is particularly noticeable when compared with the text between Tables 1 and 2, which gives perfectly reasonable values with a maximum of two significant digits after the period. This is even more unacceptable in Table 4, which gives numbers with 9 digits after a period. Are the authors willing to meaningfully interpret these strange numbers from the ROC analysis?
In my opinion, references in the list of references should be unified. Either all authors shoul be indicated by full names, or only surnames with initials. I have to recommend to check the rules of the journal.
Finally, it seems to me that the Conclusions section should be written more specifically, indicating which patients need special postoperative follow-up.
Author Response
Dear Reviewer,
We would like to thank you for taking the necessary time and effort to review our manuscript. We sincerely appreciate all your valuable comments and suggestions, which have significantly contributed to improving the quality and clarity of our manuscript. Please, find our replies to your comments below.
Comments and Suggestions for Authors
The fact that low level of coronary bypass flow reserve (CBFR) should be a most informative predictor of graft failure seems to be no wonder. However, the experimental confirmation of this self-evident effect is of independent value. The authors demonstrated that the level of CBRF which may be obtained during the operation permits assessment of the possibility of graft failure due to microvascular disorders. This knowledge may be very useful for further treatment of operated patients.
However, when reading the manuscript, a number of questions and, accordingly, critical remarks arise.
Firstly, to assess coronary reserve, the authors administered intravenous ATP, which relaxes smooth muscle in an endothelium-dependent manner. Naturally, the resulting increase in coronary blood flow is determined not only by the tone and patency of microvessels, but also by the state of the endothelium of the entire vascular bed, including relatively large coronary arteries. It is unclear how to distinguish the ability to relax the smooth muscles of microvessels against this background.
Reply: Microvascular dysfunction (MVD) describes a varied set of conditions which includes vessel destruction, abnormal vasoreactivity, in situ thrombosis, and fibrosis which ultimately results in tissue damage and progressive organ failure. Unifying hypothesis suggests that microvascular dysfunction of specific organs is an expression of a systemic illness that worsens with age and is accelerated by vascular risk factors. Endothelial dysfunction, capillary rarefaction, microcirculatory plugging with microthrombi and microemboli, microvascular remodeling, and impaired autoregulation are key pathophysiologic mechanisms shared amongst MVD in different organs. We fully agree with the statement that the increase in coronary blood flow that occurs during pharmacological stress load is determined not only by the tone and patency of microvessels, but also by the state of the endothelium of the entire vascular bed, including not only relatively large coronary arteries, but also coronary bypass grafts. And the response to such a pharmacological stress will always be systemic, where microcirculation plays the most important role. We think that an isolated assessment of the effect of ATP on microvessels, excluding the effect on larger-caliber vessels, is not possible, since this cannot be accompanied by isolated vasodilation of microvessels. And according to modern literature, сoronary microvascular dysfunction is defined as limited coronary flow reserve and/or endothelial dysfunction that contributes to myocardial ischemia and angina without interpretation of coronary artery caliber
(Minor point: I should also note here that 160 ug/kg/min is not a dose, but a rate of administration, which can be converted into a dose when multiplied by the infusion time (4 min).)
Reply: We completely agree. The text has been corrected.
Secondly, in Tables 1 and 2, the number of patients (N=238) should be removed to the legend, since the appearance of this number in the tables violates the dimensions of the rows and is completely inappropriate.
Reply: The tables have been corrected.
In these tables, it is also necessary to analyze the numbers that characterize mean and standard deviation. These values obtained during calculations cannot have so many significant digits after the decimal point. This meaninglessness is particularly noticeable when compared with the text between Tables 1 and 2, which gives perfectly reasonable values with a maximum of two significant digits after the period. This is even more unacceptable in Table 4, which gives numbers with 9 digits after a period. Are the authors willing to meaningfully interpret these strange numbers from the ROC analysis? –
Reply: We completely agree. In the tables, all digits have been reduced to two after the decimal point.
In my opinion, references in the list of references should be unified. Either all authors should be indicated by full names, or only surnames with initials. I have to recommend to check the rules of the journal.
Reply: Links are unified. Surnames with initials are indicated
Finally, it seems to me that the Conclusions section should be written more specifically, indicating which patients need special postoperative follow-up.
Reply: The conclusion of the section now indicates which patients require special postoperative monitoring. And a study limitations section has also been added
Reviewer 2 Report
Comments and Suggestions for Authors
The paper deals with the interesting topic of intraoperative bypass graft evaluation through a dynamic flow index.
The overall merit of the argumentation should be high because a myocardial perfusion evaluation at baseline and immediately after surgical intervention could be very useful in establishing the effective success of coronary revascularization. However, the data presentation and statistical analysis supporting the conclusions are misleading. In my opinion, the following sections have to be improved:
1- A table with a demographic description of the study population is lacking. If the authors intend to argue about surgical graft durability, all cardiovascular risk factors (hypertension, diabetes, history of CAD, dyslipidemia) should be reported and considered.
2- The graft failure overall events number is not displayed in the text. Furthermore, the univariable analysis for the prediction of graft failure is not reported. A multivariable analysis cannot be supported without a preliminary univariable analysis. The analysis in its current form shows that a choice of variables seems to be made on an arbitrary basis.
3- The ROC analysis is well conducted. A cut-off value for CBFR has the potential to predict the long-term outcome in patients who underwent bypass surgery. Nevertheless, this dynamic index has to be validated in a wider patient cohort and the presented results represent only a preliminary indication. A "Limitations" section is currently lacking in the paper.
Comments on the Quality of English LanguageThere are no major concerns about the English form.
Author Response
Dear Reviewer,
We would like to thank you for taking the necessary time and effort to review our manuscript.
We sincerely appreciate all your valuable comments and suggestions, which have significantly
contributed to improving the quality and clarity of our manuscript.
Please, find our replies to your comments below.
Comments and Suggestions for Authors
The paper deals with the interesting topic of intraoperative bypass graft evaluation through a dynamic flow index.
The overall merit of the argumentation should be high because a myocardial perfusion evaluation at baseline and immediately after surgical intervention could be very useful in establishing the effective success of coronary revascularization. However, the data presentation and statistical analysis supporting the conclusions are misleading. In my opinion, the following sections have to be improved:
1- A table with a demographic description of the study population is lacking. If the authors intend to argue about surgical graft durability, all cardiovascular risk factors (hypertension, diabetes, history of CAD, dyslipidemia) should be reported and considered.
Demographic data table added
2- The graft failure overall events number is not displayed in the text. Furthermore, the univariable analysis for the prediction of graft failure is not reported. A multivariable analysis cannot be supported without a preliminary univariable analysis. The analysis in its current form shows that a choice of variables seems to be made on an arbitrary basis.
Initially, a univariate logistic regression analysis was carried out for each of the potential predictor variables of graft failure. Variables significant at p < 0.05 were also included in a multivariate regression model. The following parameters were included as categorical indicators: hypertension, family history of cardiovascular disease (CVD), dyslipidemia, diabetes mellitus, hypertension, target coronary artery, type of selected conduit, previous myocardial infarction. The quantitative indicators included the results of coronary blood flow velocity characteristics according to dynamic SPECT data in the preoperative period, diameter and degree of stenosis of the coronary arteries and parameters of dynamic intraoperative ultrasound flowmetry of coronary bypass grafts (TTFM) during surgery. Variables that were significant at p < 0.05 were also included in a multivariate regression model. A mention of a univariate logistic regression analysis has been added to the text of the article.
3- The ROC analysis is well conducted. A cut-off value for CBFR has the potential to predict the long-term outcome in patients who underwent bypass surgery. Nevertheless, this dynamic index has to be validated in a wider patient cohort and the presented results represent only a preliminary indication. A "Limitations" section is currently lacking in the paper.
We fully agree that CBFR should be tested in a larger cohort of patients, which we continue to recruit. The results presented are indeed only preliminary indications, but they have reached statistical significance. We have additionally noted this in the Study Limitations section.
Comments on the Quality of English Language
There are no major concerns about the English form.
Reviewer 3 Report
Comments and Suggestions for Authors
Kozlov et. al. investigate several fachtor, which may act as predictors of graft failure after coronay bpass surgery and come to the conclusion that coronary flow reserve and the mean graft flow may predict the graft failure aftzer the bypass surgery.
- No data on at least patients' sex and age distributions is provided.
- Other important information on the patiens are also needed, e.g. do patients have other cardiac complication such as cardiac hyperterophy?
- Please describe the legend for the Figuure 1 in more detail.
- What are the correlations (r2) between the parameters?
Comments on the Quality of English Language
Minor editing required.
Author Response
Dear Reviewer,
We would like to thank you for taking the necessary time and effort to review our manuscript. We sincerely appreciate all your valuable comments and suggestions, which have significantly contributed to improving the quality and clarity of our manuscript. Please, find our replies to your comments below.
No data on at least patients sex and age distributions is provided
Demographic data table added
- Other important information on the patients are also needed, e.g. do patients have other cardiac complications such as cardiac hypertrophy?
There was no cardiac hypertrophy in the studied patients.
- - Please describe the legend for the Figure 1 in more detail
Legend for the Figure 1 has been added
- - What are the correlation (r2) between the parameters?
In this study initially, a univariate logistic regression analysis was carried out for each of the potential predictor variables of graft failure. Variables significant at p < 0.05 were also included in a multivariate regression model. Correlation (r2) between the parameters was not carried out
Round 2
Reviewer 2 Report
Comments and Suggestions for Authors
The paper has been correctly updated.
In consideration of the argumentation and the author's answers, the title might be changed to "Intraoperative prediction of coronary graft failure based on transit time flow measurement: "A PRELIMINARY STUDY"